# Stimulus-Responsive, Gelatin-Containing Supramolecular Nanofibers as Switchable 3D Microenvironments for Cells

**DOI:** 10.3390/polym14204407

**Published:** 2022-10-19

**Authors:** Kentaro Hayashi, Mami Matsuda, Masaki Nakahata, Yoshinori Takashima, Motomu Tanaka

**Affiliations:** 1Center for Integrative Medicine and Physics, Institute for Advanced Study, Kyoto University, Kyoto 606-8501, Japan; 2Department of Macromolecular Science, Graduate School of Science, Osaka University, Osaka 560-0043, Japan; 3Institute for Advanced Co-Creation Studies, Osaka University, Osaka 565-0871, Japan; 4Physical Chemistry of Biosystems, Institute of Physical Chemistry, Heidelberg University, 69120 Heidelberg, Germany

**Keywords:** gelatin nanofiber, electrospinning, supramolecular crosslink, in situ AFM nano-indentation, elasticity switching

## Abstract

Polymer- and/or protein-based nanofibers that promote stable cell adhesion have drawn increasing attention as well-defined models of the extracellular matrix. In this study, we fabricated two classes of stimulus-responsive fibers containing gelatin and supramolecular crosslinks to emulate the dynamic cellular microenvironment in vivo. Gelatin enabled cells to adhere without additional surface functionalization, while supramolecular crosslinks allowed for the reversible switching of the Young’s modulus through changes in the concentration of guest molecules in culture media. The first class of nanofibers was prepared by coupling the host–guest inclusion complex to gelatin before electrospinning (pre-conjugation), while the second class of nanofibers was fabricated by coupling gelatin to polyacrylamide functionalized with host or guest moieties, followed by conjugation in the electrospinning solution (post-conjugation). In situ AFM nano-indentation demonstrated the reversible switching of the Young’s modulus between 2–3 kPa and 0.2–0.3 kPa under physiological conditions by adding/removing soluble guest molecules. As the concentration of additives does not affect cell viability, the supramolecular fibers established in this study are a promising candidate for various biomedical applications, such as standardized three-dimensional culture matrices for somatic cells and the regulation of stem cell differentiation.

## 1. Introduction

Tissue homeostasis in multicellular organisms is sustained by the continuous remodeling of cells and extracellular matrix (ECM). Proteolytic degradation of ECM, such as the digestion of fibrous collagen by metalloprotease, enables cancer cells to invasively migrate into tissues [1,2]. In the case of muscle damage, the accumulation of fibrous type I collagen near the damage leads to an increase in ECM elasticity, which activates muscle regeneration through the proliferation of stem cells [3]. To date, matching the mechanical properties of cells and ECM, known as mechano-compliance, has been modeled using hydrogels that exhibit the elasticity of ECM [4]. However, an increasing number of studies have shown that the behavior of cells on two-dimensional (2D) substrates is distinctly different from that on three-dimensional (3D) ECMs in vivo. Moreover, type I collagen and fibronectin, two major classes of ECM proteins, are fibrous and form “mesh-like” 3D microenvironments [5,6,7].

Polymer- and/or protein-based nanofibers are considered well-defined models of natural 3D ECMs [8,9,10]. Gelatin, a hydrolysate of collagen, has been widely applied as a biomaterial [11]. Chemically crosslinked gelatin fibers facilitate the proliferation of osteocarcinoma cells [12] and the maintenance of human-induced pluripotent stem (hiPS) cells [13]. Yu et al. reported that the adhesion of hiPS cells to nanofiber-coated substrates is weaker than that to Matrigel [14]. Intriguingly, hiPS cells on nanofibers can be categorized into two sub-groups, weakly and strongly adhering cells, of which the former show a higher level of pluripotency than the latter. By adjusting the mechanical properties of synthetic fibers to match those of endogenous fibrous ECM, it is possible to discriminate against the differential invasion of cancer cells [15] and to promote certain cellular motions, such as discontinuous hopping induced by bending fibrous ECM [16]. However, the micromechanical environments of cells in vivo are never homogeneous or static, especially during highly dynamic processes, such as development and disease progression. Significant structural and mechanical remodeling of ECM is associated with various diseases, such as the proteolytic digestion of elastin caused by chronic obstructive pulmonary disease (COPD) [17] and the stiffening of bone marrow caused by blood cancer [18]. Palmquist et al. recently showed that cells actively change the alignment of fibronectin fibers and undergo collective migration during the formation of follicle structures in chick embryos [19].

These studies indicate that there is a demand for biocompatible fibrous materials with mechanical properties that respond to external stimuli in order to ensure cell viability. Previously, we prepared hydrogels of physically crosslinked micelles of triblock copolymers with pH-responsive blocks, and we showed that the morphology and adhesion strength of cells to this hydrogel can be reversibly changed by adjusting the mechanical properties of the hydrogel through changes in the pH [20,21]. Later, to avoid pH-induced changes, we fabricated hydrogels by modifying the acrylamide monomer with β-cyclodextrin (βCD) as the host and adamantane (Ad) as the guest to crosslink the polyacrylamide hydrogel with host–guest interactions. Supramolecular crosslinks enable dynamic adjustment of the Young’s modulus of fibers because the number of host–guest complexes changes in response to the concentration of host–guest molecules in the culture medium. Our data showed that the morphology of cells on the surface can be reversibly switched by stiffening or softening the hydrogel substrate [22,23,24]. More recently, we synthesized polyacrylamide-based supramolecular hydrogels functionalized with gelatin side chains in addition to host (βCD) and guest (Ad) moieties in order to circumvent tedious surface functionalization [25].

In this study, we fabricate two types of stimulus-responsive, gelatin-containing supramolecular nanofibers by electrospinning (Figure 1). The first fabrication strategy involves the coupling of the βCD/Ad inclusion complex to gelatin side chains, and thus host and guest moieties are pre-conjugated (Method 1). The second fabrication strategy aims to functionalize polyacrylamide chains containing either βCD or Ad with gelatin, and thus βCD-gelatin and Ad-gelatin are conjugated after the synthesis (post-conjugation, Method 2). Integrating gelatin allows for the direct coupling of cells without additional surface functionalization. To achieve sufficient stability under physiological conditions and to enable elasticity switching, we systematically vary the degree of chemical crosslinking and monitor the Young’s modulus in situ by indenting the fibers with an atomic force microscopy (AFM) cantilever coupled to a colloidal particle during the exchange of buffer with and without 5 mM Ad-COONa.

## 2. Materials and Methods

### 2.1. Materials

Phosphate-buffered saline (PBS) (137 mM NaCl, 8.1 mM Na_2_HPO_4_, 2.68 mM KCl, 1.47 mM KH_2_PO_4_, pH 7.4), toluene, acetone, dimethyl sulfoxide (DMSO), 1-adamantanamine (Amino-Ad), and D_2_O were purchased from Wako Pure Chemical Industries (Osaka, Japan). Ethanol was purchased from Shinwa Alcohol Industry (Tokyo, Japan). Mono-(6-amino-6-deoxy)-β-cyclodextrin (Amino–βCD) was prepared following a reported procedure [26]. Mono-6-(deoxy-acrylamido)-β-cyclodextrin (βCD–AAm) and adamantane–acrylamide (Ad–AAm) were obtained from Yushiro Chemical Industry (Tokyo, Japan). Sodium hydroxide, acrylamide (AAm), 3-aminopropyltriethoxysilane (APTES), *N*-hydroxysuccinimide (NHS), lithium bromide (LiBr), and 2-(*N*-morpholino)ethane sulfonic acid (MES) were purchased from Nacalai Tesque (Kyoto, Japan). Lithium phenyl (2,4,6-trimethylbenzoyl) phosphinate (LAP), 1-(3-dimethylaminopropyl)-3-ethylcarbodiimide hydrochloride (EDC), and 1-adamantanecarboxylic acid were purchased from Tokyo Chemical Industry (Tokyo, Japan). Gelatin type A from porcine skin (bloom strength of ~300), gelatin type B from bovine skin (bloom strength of ~225), and methacrylic anhydride were purchased from Sigma–Aldrich (Tokyo, Japan). Water used to prepare aqueous solutions was purified using a Millipore Integral MT system (Tokyo, Japan). Unless otherwise stated, these reagents were used without further purification.

### 2.2. Synthesis

#### 2.2.1. Pre-Conjugated Gelatin-βCD-Ad (Method 1)

Figure 2 shows the synthesis of pre-conjugated gelatin-βCD-Ad by the coupling of inclusion complex amino-βCD/amino-Ad and gelatin. It should be noted that the synthesis of amino-βCD was previously reported [27] but not the inclusion complex (amino-βCD/amino-Ad).

First, we prepared inclusion complex amino-βCD/amino-Ad. Amino-βCD (3.4 g, 3 mmol) and amino-Ad (0.45 g, 3 mmol) were added to water (75 mL) and stirred at 90 °C for 3 h. After cooling to room temperature, the mixture was gravity-filtered at first to remove insoluble monomers and further filtered with a syringe filter (pore size of 0.20 µm). The filtrate was freeze-dried to obtain inclusion complex amino-βCD/amino-Ad (yield of 3.2 g, 84%). The successful preparation of amino-βCD/amino-Ad was confirmed via ^1^H-^1^H 2D rotating-frame nuclear Overhauser effect spectroscopy (ROESY) NMR (Appendix A).

Next, the inclusion complex was coupled to gelatin. The exact composition of gelatin-βCD-Ad is summarized in Appendix A. Gelatin type B (2.0 g) was dissolved in MES buffer (40 mL, pH 3.7, 0.055 M) at 60 °C to obtain a transparent solution (5 *w*/*v*%). Amino-βCD/Amino-Ad complex (2.8 g, 2.2 mmol) was added to the gelatin solution. EDC (1.5 g, 8.0 mmol) and NHS (0.92 g, 8.0 mmol) were dissolved in the buffer solution at 30 °C and stirred for 18 h. To remove unreacted compounds, the reaction mixture was poured into a dialysis tube, which was immersed in water (2 L). The water was exchanged six times every other day. After the dialysis, the solution was freeze-dried to obtain gelatin-βCD-Ad as a powder (yield of 3.1 g, 64%). The successful modification was confirmed by ^1^H NMR (see the Results Section and Figure 1).

#### 2.2.2. Post-Conjugation of βCD-Gelatin and Ad-Gelatin (Method 2)

In Method 2, two building blocks, βCD-gelatin and Ad-gelatin, were synthesized (Figure 3) and then conjugated. Although Rekharsky and Inoue reported that the lack of spacers connected to βCD and Ad groups could lower the affinity [28], we did not insert polymer spacers between the main chain and host–guest moieties in this study because our previous studies have shown that the influence is not significant [29,30].

βCD-gelatin was synthesized using the exact composition summarized in Appendix A. Gelatin was modified with methacrylamide, because the reactivity of methacrylamide is higher than that of acrylamide, and the presence of a methyl group makes it easier to determine the ratio of the methacrylamide unit on gelatin characterized by ^1^H NMR (see the Results Section and Figure 2). Methacrylamide-modified gelatin was dissolved in DMSO (5 mL) at 60 °C to obtain a transparent solution (0.5 *w*/*v*%). AAm (99 mol%) and βCD–AAm (1 mol%) were dissolved in the DMSO solution of methacrylamide-modified gelatin at 60 °C. The total concentration of βCD-AAm and AAm was set to 2 M. After dissolving all monomers, LAP was added to the DMSO solution of methacrylamide-modified gelatin. The concentration of LAP in the monomer solution was 0.001 M. Free-radical copolymerization was initiated by UV irradiation (λ ≈ 365 nm, 1.4 mW/cm^2^) for 2 h using a high-pressure Hg lamp (HLR100T-2, Sen Lights, Osaka, Japan) in a conical tube, resulting in βCD-gelatin. After the polymerization, the solution was diluted with water (4 mL). To remove residual monomer LAP and DMSO, methanol (40 mL) was poured into the aqueous solution to precipitate crude βCD-gelatin. To improve the purity of βCD-gelatin, precipitation was repeated 3 times. Finally, the precipitate of βCD-gelatin was dried in vacuo at 60 °C (yield of 0.72 g, 86%).

The other building block, Ad-gelatin, was synthesized following a similar procedure. The exact composition is summarized in Appendix A. Methacrylamide-modified gelatin was dissolved in DMSO (5 mL) at 60 °C to obtain a transparent solution (0.5 *w*/*v*%). AAm (99 mol%) and Ad–AAm (1 mol%) were dissolved in the DMSO solution of methacrylamide-modified gelatin at 60 °C. The total concentration of Ad–AAm and AAm was set to 2 M. After dissolving all monomers, LAP was added to the DMSO solution of methacrylamide-modified gelatin. The concentration of LAP in the monomer solution was 0.001 M. Free-radical copolymerization was initiated by UV irradiation for 2 h using a high-pressure Hg lamp (HLR100T-2, Sen Lights, Osaka, Japan) in a conical tube, resulting in the solution of Ad-gelatin. After the polymerization, the solution was diluted with water (4 mL). To remove residual monomer LAP and DMSO, methanol (40 mL) was poured into the aqueous solution to precipitate crude Ad-gelatin. To improve the purity of Ad-gelatin, precipitation was repeated 3 times. Finally, the precipitate of Ad-gelatin was dried in vacuo at 60 °C (yield of 0.67 g, 90%).

### 2.3. Gel Permeation Chromatography (GPC)

Gel permeation chromatography (GPC) was used to determine the number-average molecular weight (*M*_n_), weight-average molecular weight (*M*_w_), and molecular weight distribution (*Ð*, *M*_w_/*M*_n_) [31,32]. Chromatograms were measured at 25 °C using an EcoSEC^®^ system (HLC-8320, TOSOH, Tokyo, Japan) equipped with a TSKgel guard column (SuperAW-L, TOSOH) and a refractive index (RI) detector. The eluent was DMSO and LiBr (1.05 g/L), and the flow rate was 0.40 mL/min. The polymer sample was dissolved in the eluent prior to loading. The molecular weight of the sample was calculated with a calibration curve prepared using polyethylene glycol standards.

### 2.4. Electrospinning of Gelatin-Containing Fibers

#### 2.4.1. Electrospinning of Gelatin-βCD-Ad Fibers (Method 1)

A round glass substrate (Ø = 25 mm) was treated with plasma (air, 30 s). The treated substrate was immersed in a 0.5 vol% solution of APTES in toluene [33]. The glass substrate in a container was shaken at 500 rpm for 1 h at 50 °C. After sequential rinsing in toluene, ethanol, and deionized water, the amino-silanized glass substrate was dried at 70 °C for 18 h in air. Gelatin-βCD-Ad (22 *w*/*v*%) was dissolved in a mixed solution of acetic acid, ethyl acetate, and water (acetic acid: ethyl acetate:water ratio of 21:14:10 (*v*/*v*/*v*)) for 18 h. The gelatin-βCD-Ad solution was added to a syringe attached to a pump. Nanofibers of gelatin-βCD-Ad were generated by electrospinning (voltage of 15 kV, flow rate of 0.2 mL/h) with a spinneret (NANON-03, MECC, Fukuoka, Japan). An amino-silanized glass substrate on aluminum foil was placed 10.5 cm below the tip of the needle to collect the nanofibers. All experiments were carried out at room temperature under low humidity (<30%). After electrospinning, the nanofibers were crosslinked by immersion in an ethanol solution of EDC and NHS (12.5, 25.0, and 37.5 mM) [13,34] for 4 h. After the reaction, the nanofibers were rinsed with 70% ethanol three times and then dried.

#### 2.4.2. Electrospinning of βCD-Gelatin/Ad-Gelatin Fibers (Method 2)

βCD-gelatin and Ad-gelatin were dissolved in a mixed solution of acetic acid, ethyl acetate, and water (acetic acid: ethyl acetate:water ratio of 21:14:10 (*v*/*v*/*v*)) with a kneader (ARE-310, Thinky, Tokyo, Japan) at 2000 rpm for 40 min. The solutions of βCD-gelatin and Ad-gelatin were mixed with a kneader (2000 rpm, 3 min) at a weight ratio of 1:1 (βCD-gelatin: Ad-gelatin). An 8 wt% solution of βCD-gelatin/Ad-gelatin was prepared from the 10 wt% solution using an acid solution. The βCD-gelatin/Ad-gelatin solution was added to a syringe attached to a pump. βCD-gelatin/Ad-gelatin nanofibers were generated by electrospinning onto APTES-coated glass substrates under the same conditions as those used for gelatin-βCD-Ad fibers. After electrospinning, the nanofibers were crosslinked by immersion in an ethanol solution of EDC and NHS (0.1 and 0.2 M) for 4 h. After the reaction, the nanofibers were rinsed with 70% ethanol three times and then dried.

### 2.5. Microscopic Imaging of Electrospun Nanofibers

The electrospun fibers were imaged using an EVOS FL microscope (Life Technologies, Carlsbad, CA, USA) equipped with a 40× objective (N.A. = 0.6) and a CKX41 inverted microscope (Olympus, Tokyo, Japan) with CAch N 10× (N.A. = 0.25) and LCAch N 20× (N.A. = 0.4) objectives.

### 2.6. AFM Nano-Indentation

AFM measurements were performed using a NanoWizard 3 AFM (JPK, Berlin, Germany). Silicon nitride quadratic pyramidal tips (TAP-150Al, BudgetSensors, Sofia, Bulgaria) and borosilicate spherical tips (CP-qp-CONT104 BSG A, NanoAndMore, Wetzlar, Germany) had nominal vertical spring constants of 5 N/m and 0.1 N/m, respectively. The tips were used in contact mode in air and PBS at 25 °C. We used the thermal noise method to determine the spring constant of the cantilevers. Time-course measurements were performed with a peristaltic pump (205CA, Watson-Marlow, Buckinghamshire, UK) at a flow rate of 0.5 mL/min. The measured force–distance curves were analyzed using the Hertz model for spherical indenters [35]. The effective elastic moduli presented were obtained from *N* > 3 experiments.

## 3. Results and Discussion

### 3.1. Characterization of Chemical Components of Gelatin-βCD-Ad

The βCD content of gelatin-βCD-Ad was calculated by integrating ^1^H NMR peaks (Figure 1) [25]. Maleic acid was set as the internal standard (*), and the integral value of the peak at 6.2 ppm corresponding to maleic acid was used to normalize other integral values. Maleic acid (0.22 mg, 0.0019 mmol) and gelatin-βCD-Ad (13 mg) were dissolved in D_2_O (0.85 mL). By comparing the gelatin-βCD-Ad spectrum with the gelatin type B spectrum, the peaks of βCD introduced into gelatin-βCD-Ad were confirmed, as shown in Figure 1a. The theoretical integral value of 2H of the vinyl group of maleic acid was set to 1. The integral value of C^1^*H* (theoretical value: 7H) of βCD in gelatin-βCD-Ad (C^1^*H*, 4.95 ppm) was 5.2. The integral value of one C^1^*H* of βCD was approximately 1.48 times larger than that of 1H of the vinyl group of maleic acid. These results suggest that βCD units (0.0028 mmol) were introduced into gelatin-βCD-Ad (13 mg) at a ratio of 0.22 mmol/g (βCD/gelatin-βCD-Ad). The successful coupling of the inclusion complex amino-βCD/amino-Ad and gelatin was also verified by systematically comparing the attenuated total reflectance–Fourier transform infrared (ATR–FTIR) spectra of gelatin-βCD-Ad, gelatin type B, amino-βCD, and amino-Ad (Appendix A) [36].

### 3.2. Characterization of Chemical Components of βCD-Gelatin and Ad-Gelatin

The ratio of the functional groups introduced into βCD-gelatin and Ad-gelatin was determined from ^1^H NMR spectra (Figure 2). According to the ^1^H NMR spectra, methacrylamide-modified gelatin (0.032 mol%) was introduced into βCD-gelatin and Ad-gelatin. Additionally, βCD and Ad units (1 mol%) were introduced into βCD-gelatin and Ad-gelatin in a stoichiometric ratio. The coupling of the inclusion complex amino-βCD/amino-Ad and gelatin was verified by systematically comparing the ATR-FTIR spectra of βCD-gelatin, Ad-gelatin, methacrylamide modified gelatin, acrylamide, βCD-AAm, and Ad-AAm (Appendix A).

### 3.3. Optical Microscopy Images of Gelatin-βCD-Ad Fibers (Method 1)

Figure 3a shows a bright field microscopy image of the gelatin-βCD-Ad fibers in air after electrospinning. Continuous and uniform fibers were fabricated by adjusting the viscosity of the solution to approximately 0.8–1.0 Pas [9,37]. As shown in Figure 3b, the fibers remained uniform after chemical crosslinking in the ethanol solution of [EDC] = [NHS] = 12.5 mM for 4 h. To ensure the stability of the fibers, the samples were rinsed with ethanol, dried in air, and soaked in PBS at 37 °C for 48 h before the imaging. To determine the optimal degree of chemical crosslinking, we prepared fibers in [EDC] = [NHS] = 25.0 mM (Figure 3c) and 37.5 mM (Figure 3d) and confirmed that continuous fibers were formed at all crosslinker concentrations.

### 3.4. Topography and Mechanical Properties of Pre-Conjugated Gelatin-βCD (Method 1)

The mechanical properties of the gelatin-βCD-Ad fibers in PBS were characterized by indenting individual fibers with an AFM cantilever equipped with a SiO_2_ particle tip (radius 5 µm). As schematically presented in Figure 4a, we first performed a topological scan to find the center of the fibers. This process is necessary for indenting the fiber while avoiding the underlying substrate because the diameter of the fibers is smaller than the particle radius. Prior to the experiments in PBS, the quality of the fiber samples was checked with an AFM scan in air. The thickness of the nanofibers was within 0.1–0.2 µm in all cases (data not shown), confirming that we produced gelatin-βCD-Ad fibers in a reproducible manner. Figure 4b shows the topographic profiles of the gelatin-βCD-Ad fibers crosslinked in [EDC] = [NHS] = 12.5 mM, 25.0 mM, and 37.5 mM, measured in PBS, confirming the formation of continuous fibers with a uniform thickness under each preparative condition. The line profiles extracted from the lines in Figure 4b are shown in Figure 4c. The thickness of the gelatin-βCD-Ad fibers was in the range of 0.8–1.2 µm irrespective of the concentrations of EDC and NHS, suggesting that the degree of swelling was not significantly different among these three conditions. It should be noted that the width of the fibers suggested by the line profiles, 8–10 µm, was approximately an order of magnitude larger than the thickness. This apparent discrepancy can be explained by the overestimation of the lateral object size by scanning with a probe radius larger than the fiber diameter (Figure 4a). The relationship among the real length scale (fiber width) *W*, probe radius (5 µm) *R*, and full width at half maximum (FWHM) obtained from the scan (Figure 4c) can be expressed as follows:(1)FWHM=2WR+W24

The thickness and the corrected values for the width of the fibers are summarized in Table 1. Although the width was slightly larger than the thickness due to dissipation caused by electrospinning, chemical crosslinking, and drying, the obtained data demonstrated that cylindrical nanofibers were formed.

Figure 5a–c show the typical force–distance curves of the gelatin-βCD-Ad fibers crosslinked in [EDC] = [NHS] = 12.5 mM, 25.0 mM, and 37.5 mM, measured in PBS at 25 °C. The solid lines represent the best fit results obtained using the Hertz model for a spherical indenter (Johnson 1985):(2)$$F=\frac{4E\sqrt{R}}{3\left(1-\nu^{2}\right)}\delta^{\frac{3}{2}}$$

The Young’s moduli *E* obtained with four independent measurements are summarized in Table 1. The Young’s modulus exhibited a monotonic increase with an increase in EDC/NHS, suggesting that a higher degree of chemical crosslinking was achieved at a higher concentrations of EDC and NHS. These values are in reasonable agreement with the Young’s moduli of chemically crosslinked gelatin nanofibers previously reported [10,15]. In the next step, we examined if the addition of competitive host–guest molecules modulates the Young’s modulus by freeing supramolecular crosslinks. As the additive, we chose 5 mM Ad-COONa, which was used in our previous study to modulate the Young’s modulus of βCD-Ad-gelatin hydrogels without interfering with cell viability [25]. For the gelatin-βCD-Ad fibers prepared at different concentrations of EDC and NHS, the Young’s moduli measured in the absence and presence of 10 mM Ad-COONa (Figure 5d) were not significantly different. These data suggest that the change caused by competitive Ad-COONa is counteracted by stable crosslinks, such as chemical crosslinking by EDC and NHS or the physical entanglement of gelatin. We examined the former scenario by decreasing the concentrations ofEDC and NHS. However, the fibers crosslinked in [EDC] = [NHS] = 6.25 mM were not stable in PBS. Therefore, we used the second strategy (Method 2 in Figure 1): the post-conjugation of gelatin-functionalized polyacrylamide chains with βCD and Ad side chains.

### 3.5. Topography and Mechanical Properties of Post-Conjugated βCD-Gelatin and Ad-Gelatin (Method 2)

Figure 6a shows the topographic profile of the post-conjugated βCD-gelatin/Ad-gelatin fibers measured in air, revealing that continuous fibers of uniform thickness were produced when the viscosity of the polymer solution was adjusted to approximately 0.8–8.1 Pas. The thickness and width of the fibers were within the range of 0.3–0.5 µm and 2.4–3.6 µm, respectively. The electrospun fibers were crosslinked in solution EDC and NHS for 4 h, rinsed with ethanol, dried in air, and soaked in PBS at 25 °C for 2 h. In contrast to the gelatin-βCD-Ad fibers, the βCD-gelatin/Ad-gelatin fibers required highly concentrated solutions of EDC and NHS for chemical crosslinking. As shown in Figure 6b, fibers treated with [EDC] = [NHS] = 100 mM exhibited pearl-like features, indicating that the fibers were not stable enough to sustain their original cylindrical structures. In fact, we did not find any fibers on the substrate when the fibers were treated with [EDC] = [NHS] < 100 mM [37]. Stable fibers were only found at [EDC] = [NHS] ≥ 200 mM, as shown in Figure 6c–e. The thickness *D* and width *W* of βCD-gelatin/Ad-gelatin fibers calculated using Equation (1) are summarized in Table 2. Although the values were slightly larger than the corresponding values of the pre-conjugated gelatin-βCD-Ad fibers (Table 1), we verified that the post-conjugated βCD-gelatin/Ad-gelatin fibers crosslinked in highly concentrated [EDC] = [NHS] ≥ 200 mM were stable over 48 h in buffer. The corresponding optical microscopy images are shown in Appendix A.

The Young’s modulus of the crosslinked fibers was measured using AFM nano-indentation following the same protocols used for the gelatin-βCD-Ad fibers. A characteristic force–distance curve of a βCD-gelatin/Ad-gelatin fiber crosslinked in [EDC] = [NHS] = 2 M is shown in Figure 6f. The Young’s moduli calculated using Equation (2) are summarized in Table 2. It is notable that the obtained Young’s moduli of the βCD-gelatin/Ad-gelatin fibers were one order of magnitude lower than those of the gelatin-βCD-Ad fibers (Table 1) despite the fact that the concentrations of EDC and NHS used for chemical crosslinking was much higher for the former than for the latter. The Young’s modulus was 1.6 ± 0.4 kPa, even when the concentration of EDC and NHS was close to saturation (2 M). This finding can be attributed to several reasons. The poly(acrylamide) main chain can suppress the non-specific interactions between gelatin units, such as physical entanglement, and hence reduces the Young’s modulus. Furthermore, the number of free carboxyl groups available for crosslinking is insufficient because of the low amount of gelatin in βCD-gelatin (Appendix A) and Ad-gelatin (Appendix A) [38]. It is plausible that the Young’s modulus does not increase with an increase in crosslinker concentration if free carboxyl groups are consumed. 

### 3.6. Reversible Switching of Young’s Modulus of βCD-Gelatin/Ad-Gelatin Fibers (Method 2)

The low Young’s moduli obtained for the βCD-gelatin/Ad-gelatin fibers suggest that this parameter is susceptible to change in response to chemical stimuli, such as the addition of 5 mM Ad-COONa (Figure 5). In fact, the βCD-gelatin/Ad-gelatin fibers crosslinked in [EDC] = [NHS] = 100 mM (Figure 6b) disappeared when the buffer was exchanged with PBS containing 5 mM Ad-COONa (data not shown). Here, we connected the AFM sample holder to a peristaltic pump and monitored the change in the Young’s modulus and fiber thickness while exchanging the buffer. Figure 7a,b show the change in the Young’s modulus and the thickness of the βCD-gelatin/Ad-gelatin fibers crosslinked in [EDC] = [NHS] = 400 mM over time. The gray-shaded zones (Ad) correspond to the period in which the fibers are in contact with the PBS containing 5 mM Ad-COONa, while the white zone corresponds to the period in which the fibers are in contact with the PBS with no Ad-COONa. Exchanging Ad-free buffer (white) with Ad-loaded buffer (gray) led to a rapid decrease in the Young’s modulus from *E*_Ad-free_ ≈ 2.4 kPa to *E*_Ad-loaded_ ≈ 0.2 kPa and an increase in the fiber thickness from *D*_Ad-free_ ≈ 1.7 µm to *D*_Ad-loaded_ ≈ 2.0 µm. This finding can be explained by the decrease in the density of βCD/Ad complexes owing to the presence of competitive Ad-COONa in the solution. In contrast, exchanging Ad-loaded buffer (gray) with Ad-free buffer (white) resulted in a change in the Young’s modulus and in the thickness in the opposite direction. The Young’s modulus increased to *E*_Ad-free_ ≈ 2.5 kPa, and the thickness decreased to *D*_Ad-free_ ≈ 1.5 µm. Stiffening/thinning was slower than softening/thickening, which is consistent with the results of previous studies on hydrogels crosslinked with reversible supramolecular βCD/Ad interactions [23,25].

To determine if the crosslinking at high concentrations of EDC and NHS affects the response of the fibers to external stimuli, we performed in situ AFM nano-indentation experiments with the βCD-gelatin/Ad-gelatin fibers crosslinked in [EDC] = [NHS] = 2 M. Changes in the Young’s modulus and thickness over time are presented in Figure 7c,d. Although the absolute values of the Young’s modulus and thickness at *t* = 0 min were slightly different between the fibers crosslinked in [EDC] = [NHS] = 400 mM and the fibers crosslinked in [EDC] = [NHS] = 2 M (*E*_2M,Ad-free_ ≈ 1.6 kPa and *D*_2M,Ad-free_ ≈ 1.4 µm), the βCD-gelatin/Ad-gelatin fibers exhibited the same reversible response to the exchange of buffers. The difference in the Young’s modulus for the fibers crosslinked in [EDC] = [NHS] = 2 M (Δ*E*_2M_ ≈ 1.3 kPa) was slightly smaller than that for the fibers crosslinked in [EDC] = [NHS] = 400 mM (Δ*E*_400mM_ ≈ 2.4 kPa). However, it is currently not possible to attribute this difference to either the different initial elasticity levels or the different densities of the chemical crosslinks.

Remarkably, both the βCD-gelatin/Ad-gelatin fibers exhibited reversible switching of the Young’s modulus and thickness in the physiological buffer in response to the addition and removal of 5 mM Ad-COONa.

## 4. Conclusions

In this study, we fabricated two types of stimulus-responsive, gelatin-containing supramolecular nanofibers that can be utilized as well-defined, switchable 3D microenvironments for cells. The first nanofibers were synthesized by coupling the βCD/Ad inclusion complex to gelatin (called pre-conjugation), whereas the second nanofibers were fabricated by mixing gelatin-functionalized polyacrylamide chains coupled to either βCD or Ad (called post-conjugation). The balance between supramolecular crosslinks and chemical/covalent crosslinks was optimized by varying the concentration of EDC and NHS, yielding fibers that are stable under physiological conditions. The pre-conjugated fibers exhibited a monotonic increase in the Young’s modulus from 16 kPa to 42 kPa with an increase in [EDC] = [NHS] = 12.5 mM to 37.5 mM. However, the addition of Ad-COONa solution did not cause any change in the Young’s modulus, suggesting that the change caused by competitive Ad-COONa is screened by other strong interactions, such as the physical entanglement of gelatin or a large amount of chemical crosslinks. However, the post-conjugated fibers exhibited about one order of magnitude lower *E* = 1–3 kPa, even at [EDC] = [NHS] = 2 M, which can be attributed to either the saturation of available carboxyl groups or the cancellation of gelatin entanglement by polyacrylamide chains. In situ AFM nano-indentation demonstrated the reversible switching of the Young’s modulus between *E* = 1–3 kPa and 0.2–0.3 kPa by adding/removing 5 mM Ad-COONa, which does not interfere with the viability of cells [25]. Although the fiber elasticity is about one order of magnitude lower than that of naturally occurring fibrous ECM (*E*~10 kPa) [15], further optimization of the polymer composition, such as increasing the number of carboxyl groups by copolymerization with other monomers or suppressing gelatin–gelatin interactions and integrating fibrous materials into 3D micro-scaffolds [16,24,39], may lead to well-defined 3D cellular microenvironments with switchable mechanical properties.

## Data Availability

The authors declare that the main data supporting the results in this study are available within the paper and its Appendix A. The raw datasets generated during the study are available for research purposes from the corresponding authors on reasonable request.

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
