# Peer review of "Stimulus-Responsive, Gelatin-Containing Supramolecular Nanofibers as Switchable 3D Microenvironments for Cells"

_polymers, 2022, doi:10.3390/polym14204407_

Round 1

Reviewer 1 Report

I reviewed the manuscript entitled, Stimulus Responsive, Gelatin-Containing Supramolecular Nanofibers as Switchable 3D Microenvironments for Cells. The manuscript is well written and scientifically sounds high. Research hypothesis and approach is novel and contributes to the field. In my opinion, this manuscript can be accepted for publication after considering the suggestions below.

The format of the manuscript is not appropriate in some places. Please revise according to the journal format.

Abstract

Abstract is well written; however, authors should include concluding remark and practical application of the study for real-world problems

Introduction

Line 77: Scheme 1? I think it should be Figure. Please revise such words like scheme and replace with Figure

Materials and methods

Line: 91: Wako Pure Chemical Industries… please write city and country of it

Line 95; city?

Line 111: Scheme 2 should be replaced with Figure 2. Please replace all “Scheme” with Figure

Section 2.3. provide citation for the method

Results and discussion:

Figures 6 and 7: quality must be improved

This section is well written and compared with available scientific literature.

Authors must provide the conclusion section and provide short note on concluding remarks of the study and future recommendations

References are not according to the journal format. Please revise carefully

Author Response

We thank the reviewer for the careful reviewing and constructive suggestion. In the attached file, we made point-by-point answers to the reviewer's comments.

Reviewer 2 Report

This is a well-written paper about supramolecular nanofibers as switchable 3D microenvironments for cells. I recommend it for publication after the following minor points are solved.

1. Line 30-31, several recent reviews (Pharmaceutics 2022, 14(5), 998; Advanced Materials 33 (18), 2005513, 2021; Acta Biomaterialia 132, 83-102, 2021) should be included to support such a claim.

2. There is no spacer between the polymer and CD or Ad. Does this configuration decrease the affinity between CD and Ad?

3. Why was gelatin modified with methacrylamide but CD and Ad were modified with acrylamide?

4. It would be better if the authors could add a section of the conclusion.

Author Response

(The authors gave the same response as above.)

Reviewer 3 Report

In this work, Stimulus Responsive, Gelatin-Containing Supramolecular Nanofibers as Switchable 3D Microenvironments for Cells. The idea of this research is interesting to readers. The background is well studied and the presentation of the method is very clear and sound, but there are some minor issues to be addressed:

Comments:

The author should provide some quantitative information in the abstract section.

The author should cite suitable reference in the section 2.2.1. Pre-conjugated gelatin-βCD-Ad

The author should cite important reference in the Results and discussion section

There is no separate conclusion section in this manuscript. The author should the conclusion section with important results.

Author Response

(The authors gave the same response as above.)
